# Measure of Weak Noncompactness and Fixed Point Theorems in Banach Algebras with Applications

**Mohamed Amine Farid** [1], **Karim Chaira** [2], **El Miloudi Marhrani** [1,*] and **Mohamed Aamri** [1]

1   Laboratory of Algebra, Analysis and Applications (L3A), Faculty of Sciences Ben M'Sik, Hassan II University of Casablanca, B.P 7955, Sidi Othmane, Casablanca 20700, Morocco; amine.farid17@gmail.com (M.A.F.); aamrimohamed82@gmail.com (M.A.)
2   CRMEF, Avenue Allal El Fassi, Madinat Al Irfan, B.P 6210, Rabat 10000, Morocco; chaira_karim@yahoo.fr
*   Correspondence: marhrani@gmail.com

**Abstract:** In this paper, we prove some fixed point theorems for the nonlinear operator $A \cdot B + C$ in Banach algebra. Our fixed point results are obtained under a weak topology and measure of weak noncompactness; and we give an example of the application of our results to a nonlinear integral equation in Banach algebra.

**Keywords:** Banach algebras; fixed point theorems; measure of weak noncompactness; weak topology; integral equations

## 1. Introduction

Integral equations are involved in various scientific problems such as transport theory, the theory of radiative transfer, biomathematics, etc (see [1–6]). The use of these equations dates back to 1730 with Bernoulli in the study of oscillatory problems. With the development of functional analysis, more general results were obtained by L. Schwartz, H. Poincaré, I. Fredholm, and others.

The problems of the existence of solutions for an integral equation can then be resolved by searching fixed points for nonlinear operators in a Banach algebra. For this, many researchers have been interested in the case where the Banach algebra is endowed with its strong topology; however, few of them were interested to the existence of a fixed point for mappings acting on a Banach algebra equipped with its weak topology [7–11]; such a topology allows obtaining some generalizations of these results.

The history of fixed point theory in Banach algebra started in 1977 with R.W. Legget [12], who considered the existence of solutions for the equation:

$$x = x_0 + x \cdot Bx, \ (x_0, x) \in X \times \Omega \tag{1}$$

where $\Omega$ is a nonempty, bounded, closed, and convex subset of a Banach algebra $X$ and $B$ is a compact operator from $\Omega$ into $X$. Many authors [10,11,13,14] generalized Equation (1) to the equation:

$$x = Ax \cdot Bx + Cx, \ x \in \Omega, \tag{2}$$

where $\Omega$ is a nonempty, bounded, closed, and convex subset of a Banach algebra and $A, C : X \longrightarrow X$, $B : \Omega \longrightarrow X$ are nonlinear operators. Most of these authors have obtained the desired results through the study of the operator $\left(\frac{I-C}{A}\right)^{-1} B$.

This study was based mainly on the properties of operators $A$, $B$, $C$, and $\frac{I-C}{A}$ (cf. condensing, relatively weakly compact, etc.).

The study of nonlinear integral equations in Banach algebra via fixed point theory was in initiated by B.C. Dhage [15]. In 2005, B.C. Dhage [14] studied the existence of solutions for the equation:

$$x = Ax \cdot Bx$$

The results were obtained in the case of the norm topology on Banach algebra. In 2014, Banas et al. [8] proved some existence results of operator equations under the weak topology using the measure of weak noncompactness. In 2015, Ben Amar et al. used the De Blasi measure of non-compactness to obtain some generalizations of these results. In 2019, A.B. Amar et al. [16] established new fixed point theorems for the sum of two mappings in Banach space and showed that the condition «weakly condensing» can by relaxed by the assumption «countably weakly condensing».

In this paper, we use the measure of noncompactness to prove some fixed point results for a nonlinear operator of type $AB + C$ in a Banach algebra. We note that the condition «relatively weakly compact », which is not easy to verify, is not required in most results in [16]. Our results are formulated using the operator $I - \frac{I-C}{A}$ under the weak topology in a Banach algebra.

As an application, we discuss the existence of solutions for an abstract nonlinear integral equation in the Banach algebra $C([0,1], X)$; and an example of a nonlinear integral equation in the Banach algebra $C([0,1], \mathbb{R})$.

## 2. Preliminaries

Let $(X, \| \ \|)$ be a Banach space with zero element $\theta$. We denote respectively $P(X)$, $P_{cv}(X)$, $P_{bd}(X)$ and $P_{cl,cv}(X)$ the family of all nonempty subsets, nonempty and convex subsets, nonempty and bounded subsets, nonempty closed and convex subsets of $X$.

For any $\varepsilon > 0$, we denote $B_\varepsilon$ the closed ball of $X$ centered at origin with radius $\varepsilon$. Moreover, we write $x_n \to x$ and $x_n \rightharpoonup x$ respectively to denote the strong convergence and the weak convergence of a sequence $\{x_n\}_n$ to $x$.

For a subset $K$ of $X$, we write $\overline{K}$, $\overline{K}^w$, $convK$, and $\overline{conv}K$, to denote the closure, the weak closure, the convex hull, and the closed convex hull of the subset $K$, respectively; and by $\mathcal{R}(T)$, the range of the operator $T$.

**Definition 1.** *Let $\Omega$ be a nonempty subset of $X$. We say that a multivalued map $H : \Omega \to P(\Omega)$ has a weakly closed graph if the following property holds: if for every net $\{x_\delta\}_\delta$, with $x_\delta, x \in \Omega$ such that $x_\delta \to x$ and $\{y_\delta\}_\delta$ such that $y_\delta \in Hx_\delta$, $y_\delta \to y$, then $Hx \cap S(x,y) \neq \varnothing$; here, $S(x,y) := \{\lambda y + (1 - \lambda)x \, ; \, \lambda \in [0,1]\}$.*

We say that a map $H : \Omega \to P(\Omega)$ has a $w$-weakly closed graph in $\Omega \times X$ if it has a weakly closed graph in $\Omega \times X$ with respect to the weak topology.

**Definition 2** ([9]). *Let $X$ be a Banach space. An operator $T : X \to X$ is said to be weakly sequentially continuous on $X$ if for every sequence $\{x_n\}_n$ with $x_n \rightharpoonup x$, we have $Tx_n \rightharpoonup Tx$.*

Note that $T$ is weakly sequentially continuous if and only if $I - T$ is weakly sequentially continuous.

**Definition 3.** *Let $X$ be a Banach space. An operator $T : X \longrightarrow X$ is said to be weakly compact if $T(M)$ is relatively weakly compact for every bounded subset $M \subset X$.*

**Definition 4** ([17]). *Let $\Omega$ be a nonempty weakly closed set of a Banach space $X$ and $T : \Omega \to X$ a weakly sequentially continuous operator. $T$ is said to be a weakly semi-closed operator at $\theta$ if the conditions $\{x_n\}_n \subset \Omega$, $x_n - Tx_n \to \theta$ imply that there exists $x \in \Omega$ such that $Tx = x$.*

We recall that a function $\omega : P_{bd}(X) \to [0, +\infty)$ is said to be a measure of weak noncompactness (MWNC) on $X$ if it satisfies the following properties.

1.  For any bounded subsets $\Omega_1, \Omega_2$ of $X$, we have $\Omega_1 \subseteq \Omega_2$ implies $\omega(\Omega_1) \leq \omega(\Omega_2)$.
2.  $\omega(\overline{conv}(\Omega)) = \omega(\Omega)$, for all bounded subsets $\Omega \subset X$.
3.  $\omega(\Omega \cup \{a\}) = \omega(\Omega)$ for all $a \in X, \Omega \in P_{bd}(X)$.
4.  $\omega(\Omega) = 0$ if and only if $\Omega$ is relatively weakly compact in $X$.

The MWNC $\omega$ is said to be:

1.  Positive homogeneous, if $\omega(\lambda\Omega) = \lambda\omega(\Omega)$, for all $\lambda > 0$ and $\Omega \in P_{bd}(X)$.
2.  Subadditive, if $\omega(\Omega_1 + \Omega_2) \leq \omega(\Omega_1) + \omega(\Omega_2)$, for all $\Omega_1, \Omega_2 \in P_{bd}(X)$.

As an example of MWNC, we have the De Blasi measure of weak noncompactness [18], defined on $P_{bd}(X)$ by:

$$\mu(M) = \inf\{\varepsilon > 0; \text{ there exists } K \text{ weakly compact such that} : \ M \subset K + B_\varepsilon\},$$

it is well known that $\mu$ is homogenous, subadditive, and satisfies the set additivity property:

$$\mu(M \cup N) = \max\{\mu(M), \mu(N)\}, \text{ for all } M, N \in P_{bd}(X).$$

For more properties of the MWNC, we refer to [19].

Let us formulate some other definitions needed in this paper.

**Definition 5.** *Let $\Omega$ be a subset of a Banach space $X$, $\omega$ be an MWNC on $X$, and $0 \leq k < 1$. Let $T$ be a mapping from $\Omega$ into $X$; we say that:*

1.  *$T$ is $k$-$\omega$-contractive if $\omega(T(M)) \leq k\omega(M)$ for any bounded set $M \subset \Omega$;*
2.  *$T$ is $\omega$-condensing if $\omega(T(M)) < \omega(M)$ for any bounded set $M \subset \Omega$ with $\omega(M) > 0$;*
3.  *$T$ is countably $k$-$\omega$-contractive, if $\omega(T(M)) \leq k\omega(M)$ for any countable bounded set $M \subset \Omega$;*
4.  *$T$ is countably $\omega$-condensing if $\omega(T(M)) < \omega(M)$ for any countable bounded set $M \subset \Omega$ with $\omega(M) > 0$;*
5.  *$T$ is weakly countable one-set-contractive if $\omega(T(M)) \leq \omega(M)$ for any bounded set $M \subset \Omega$.*

Clearly, every $k$-$\omega$-contractive is countably $k$-$\omega$-contractive, but the converse is not always true.

**Definition 6.** *A mapping $T : \Omega \subset X \longrightarrow X$ is said to be:*

1.  *Lipschitzian with the Lipschitz constant $k > 0$:*

$$\|Tx - Ty\| \leq k\|x - y\|, \text{ for all } x, y \in \Omega.$$

*If $k = 1$, $T$ is called nonexpansive, and if $k \in [0, 1[$, $T$ is called a contraction.*
2.  *Pseudocontractive if for each $r > 0$, we have:*

$$\|x - y\| \leq \|r(Ty - Tx) + (1 + r)(x - y)\|, \text{ for all } x, y \in \Omega.$$

3.  *Accretive if for each $\lambda \geq 0$, we have:*

$$\|x - y\| \leq \|x - y + \lambda(Tx - Ty)\|, \text{ for all } x, y \in \Omega.$$

*In addition, if $\mathcal{R}(I + \lambda T) = X$ for every $\lambda > 0$, then $T$ is called m-accretive.*

Note that $T$ is pseudocontractive if and only if $I - T$ is accretive.

**Definition 7.** *An operator* $T : \Omega \subseteq X \to X$ *is called* $\mathcal{D}$-*Lipschitzian if there exists a continuous and nondecreasing function* $\Phi_T : [0, +\infty) \to [0, +\infty)$ *with* $\Phi_T(0) = 0$ *such that:*

$$\|Tx - Ty\| \leq \Phi_T(\|x - y\|), \text{ for all } x, y \in \Omega.$$

*Sometimes,* $\Phi_T$ *is called a* $\mathcal{D}$-*function of T on X. Moreover, if* $\Phi_T(r) < r$ *for all* $r > 0$, *then the operator T is called a nonlinear contraction with a contraction function* $\Phi_T$.

**Definition 8.** *An operator* $T : \Omega \subseteq X \to X$ *is said to be* $\psi$-*expansive if there exists a function* $\psi : [0, \infty) \to [0, \infty)$ *such that* $\psi(0) = 0$, $\psi(r) > r$ *for any* $r > 0$, $\psi$ *is either continuous or nondecreasing, and* $\|Tx - Ty\| \geq \psi(\|x - y\|)$ *for all* $x, y \in \Omega$.

**Definition 9.** *We say that* $H : \Omega \subseteq X \to P(X)$ *is countably* $\omega$-*condensing if* $H(\Omega)$ *is bounded on X and* $\omega(H(M)) < \omega(M)$ *for all countable bounded subsets M of* $\Omega$ *with* $\omega(M) > 0$.

The following result is crucial:

**Theorem 1** ([20]). *Let X be a Banach space.*

(i)     *Let H be a bounded subset of* $\mathcal{C}([0, T], X)$. *Then:*

$$\sup_{t \in [0,T]} \mu(H(t)) \leq \mu(H),$$

*where* $H(t) = \{x(t); x \in H\}$.

(ii)    *Let* $H \subset \mathcal{C}([0, T], X)$ *be bounded and equicontinuous. Then:*

$$\mu(H) = \sup_{t \in [0,T]} \mu(H(t)) = \mu(H([0, T])),$$

*where* $H([0, T]) = \cup_{t \in [0,T]} H(t)$.

*Here,* $\mu$ *is the De Blasi measure of weak noncompactness.*

**Lemma 1** ([21]). *Let X be a Banach space and* $T : X \longrightarrow X$ *a k-Lipschitzian map and weakly sequentially continuous. Then, for each bounded subset S of X, we have:*

$$\mu(T(S)) \leq k\mu(S), \text{ for all } x, y \in X;$$

*here,* $\mu$ *is the De Blasi measure of weak noncompactness.*

We recall that an algebra $X$ is a vector space endowed with an internal composition law denoted by «·», which is associative and bilinear. A normed algebra is an algebra endowed with a norm $\|.\|$ satisfying the following property:

$$\|x \cdot y\| \leq \|x\|\|y\|, \text{ for all } x, y \in X.$$

A complete normed algebra is called a Banach algebra. For basic properties of Banach algebra, refer to [22].

In general, the product of two weakly sequentially continuous mappings on a Banach algebra is not necessarily weakly sequentially continuous.

**Definition 10** ([9])**.** *We will say that the Banach algebra X satisfies condition* $(\mathcal{P})$ *if:*

$$(\mathcal{P}) \begin{cases} \text{For any sequences } \{x_n\}_n \text{ and } \{y_n\}_n \text{ in } X \text{ such that } x_n \rightharpoonup x \text{ and } y_n \rightharpoonup y, \\ \text{we have } x_n y_n \rightharpoonup xy. \end{cases}$$

Note that, every finite dimensional Banach algebra satisfies condition $(\mathcal{P})$. If $X$ satisfies condition $(\mathcal{P})$, then the space $C(K; X)$ of all continuous functions from a compact Hausdorff space $K$ into $X$ is also a Banach algebra satisfying condition $(\mathcal{P})$ (see [9]).

**Definition 11.** *Let X be a Banach algebra. An operator* $T : X \to X$ *is called regular on X if it maps X into the set of all invertible elements of X.*

In [16] (Theorem 3.1), Afif Ben Amar et al. proved the following result:

**Theorem 2** ([16], Theorem 3.1)**.** *Let* $\Omega$ *be a nonempty closed convex subset of a Banach space X and* $\omega$ *be an MWNC on X. Assume that* $T : \Omega \to \Omega$ *is a weakly sequentially continuous and countably* $\omega$-*condensing mapping with a bounded range. Then, T has a fixed point.*

**Theorem 3** ([16], Theorem 3.3)**.** *Let* $\Omega$ *be a nonempty closed convex subset of a Banach space X,* $\omega$ *be a positive homogeneous MWNC on X, and* $T : \Omega \to \Omega$ *be weakly sequentially continuous, weakly countably one-set-contractive. In addition, assume that T is weakly semi-closed at* $\theta$ *with a bounded range. Then, T has a fixed point.*

**Theorem 4** ([16], Theorem 3.2)**.** *Let* $\Omega$ *be a nonempty convex closed subset of a Banach space E,* $U \subset E$ *be a weakly open subset of* $\Omega$ *with* $\theta \in U$, *and* $\omega$ *be a subadditive MWNC on E. Assume* $T : \overline{U}^w \longrightarrow X$ *is a weakly sequentially continuous countably* $\omega$-*condensing map with* $T(\overline{U}^w)$ *bounded. Then, either T has a fixed point or there exists* $u \in \partial_{\Omega} U$ *and* $\lambda \in ]0, 1[$ *such that* $u = \lambda T(u)$ *(* $\partial_{\Omega} U$ *denotes the weak boundary of U in* $\Omega$*).*

The following lemma is useful for the sequel.

**Lemma 2.** *Let X be a Banach algebra satisfying condition* $(\mathcal{P})$. *Then, for any bounded subset M of X and relatively weakly compact subset K of X, we have* $w(MK) \leq \|K\| w(M)$.

## 3. Results

Our first main result is a new version of Theorem 3.2 proven by Jeribi et al. in [23].

**Theorem 5.** *Let* $\Omega$ *be a nonempty, bounded, closed, and convex subset of a Banach algebra X and* $\omega$ *be a subadditive MWNC on X. Let* $A, C : X \longrightarrow X$, *and* $B : \Omega \longrightarrow X$ *be three operators that satisfy the following conditions:*

*(i)* $A$ *is regular on X, and* $\left(\frac{I-C}{A}\right)^{-1}$ *exists on* $B(\Omega)$,
*(ii)* $B$ *and* $\frac{I-C}{A}$ *are weakly sequentially continuous,*
*(iii)* $I - \frac{I-C}{A}$ *is countably* $\alpha$-$\omega$-*contractive on* $\Omega$,
*(iv)* $B$ *is countably* $\beta$-$\omega$-*contractive,*
*(v)* $x = Ax \cdot By + Cx$, *for all* $y \in \Omega$ *implies* $x \in \Omega$.

*Then, there exists* $x \in \Omega$ *such that* $x = Ax \cdot Bx + Cx$, *whenever* $\frac{\beta}{1-\alpha} < 1$.

**Proof.** Note that $x = Ax \cdot Bx + Cx$, $x \in \Omega$ if and only if $x$ is a fixed point for the operator $T := \left(\frac{I-C}{A}\right)^{-1} B$.

Let $y \in \Omega$; from Assumption $(i)$, there is a unique $x_y \in X$ such that:

$$\left(\frac{I-C}{A}\right) x_y = By,$$

then:

$$x_y = Ax_y \cdot By + Cx_y;$$

by Condition $(v)$, we have $x_y \in \Omega$, and then, $T$ is well defined on $\Omega$.

By Theorem 2, it suffices to prove that the map $T$ is weakly sequentially continuous and countably $\omega$-condensing.

Let $\{x_n\}_n$ be a sequence in $\Omega$ such that $x_n \rightharpoonup x$; the set $\{x_n : n \in \mathbb{N}\}$ is relatively weakly compact; and since $B$ is weakly sequentially continuous, the set $\{Bx_n : n \in \mathbb{N}\}$ is relatively weakly compact. Assume that $\omega(\{Tx_n : n \in \mathbb{N}\}) > 0$. Since:

$$T = B + \left(I - \frac{I-C}{A}\right) T,$$

and $I - \frac{I-C}{A}$ is countably $\alpha$-$\omega$-contractive, we obtain:

$$\begin{aligned}
\omega\left(\{Tx_n : n \in \mathbb{N}\}\right) &\leq \omega\left(\{Bx_n : n \in \mathbb{N}\}\right) + \omega\left(\left(I - \frac{I-C}{A}\right)(\{Tx_n : n \in \mathbb{N}\})\right) \\
&\leq \alpha\omega\left(\{Tx_n : n \in \mathbb{N}\}\right) \\
&< \omega\left(\{Tx_n : n \in \mathbb{N}\}\right),
\end{aligned}$$

which is absurd. It follows that $\{Tx_n : n \in \mathbb{N}\}$ is weakly relatively compact; hence, there exists a subsequence $\{x_{\sigma(n)}\}_n$ of $\{x_n\}_n$ such that $Tx_{\sigma(n)} \rightharpoonup y$ for some $y \in \Omega$. Moreover, $\frac{I-C}{A}$ is weakly sequentially continuous; then, $I - \frac{I-C}{A}$ is weakly sequentially continuous, and then:

$$\left(I - \frac{I-C}{A}\right) Tx_{\sigma(n)} \rightharpoonup \left(I - \frac{I-C}{A}\right) y,$$

As we have $\left(I - \frac{I-C}{A}\right) T = -B + T$ and $-Bx_{\sigma(n)} + Tx_{\sigma(n)} \rightharpoonup -Bx + y$, we obtain:

$$-Bx + y = y - \left(\frac{I-C}{A}\right) y$$

which gives $Tx = y$, and therefore, $Tx_{\sigma(n)} \rightharpoonup Tx$.

We claim that $Tx_n \rightharpoonup Tx$. Assume that there exists a subsequence $\{x_{\sigma_1(n)}\}_n$ of $\{x_n\}_n$ and a weak neighborhood $V^w$ of $Tx$ such that $Tx_{\sigma_1(n)} \notin V^w$ for all $n \in \mathbb{N}$. Since $\{x_{\sigma_1(n)}\}_n$ converge weakly to $x$, we may extract a subsequence $\{x_{\sigma_1\sigma_2(n)}\}_n$ of $\{x_{\sigma_1(n)}\}_n$ such that $Tx_{\sigma_1\sigma_2(n)} \rightharpoonup Tx$ and $Tx_{\sigma_1\sigma_2(n)} \notin V^w$, which is absurd. Hence, $Tx_n \rightharpoonup Tx$; it follows that $T$ is weakly sequentially continuous.

$T$ is countably $\omega$-condensing. Indeed, let $M$ be a countably subset of $\Omega$ with $\omega(M) > 0$; we have:

$$\begin{aligned}
\omega(T(M)) &\leq \omega(B(M)) + \omega\left(\left(I - \frac{I-C}{A}\right)(T(M))\right) \\
&\leq \beta\omega(M) + \alpha\omega(T(M)),
\end{aligned}$$

then $\omega(T(M)) \leq \frac{\beta}{1-\alpha}\omega(M) < \omega(M)$, which ends the proof. $\square$

**Corollary 1.** *Let $\Omega$ be a nonempty, bounded, closed, and convex subset of a Banach algebra $X$ and $\omega$ be a subadditive MWNC on $X$. Let $C : X \longrightarrow X$ and $B : \Omega \longrightarrow X$ be two operators that satisfy the following conditions:*

(i)   $(I - C)^{-1}$ *exists on* $B(\Omega)$,
(ii)  *B and* $I - C$ *are weakly sequentially continuous,*
(iii) *C is countably* $\alpha$-$\omega$-*contractive on* $\Omega$,
(iv)  *B is countably* $\beta$-$\omega$-*contractive,*
(v)   $x = By + Cx$, *for all* $y \in \Omega$ *implies* $x \in \Omega$.

*Then, there exists* $x \in \Omega$ *such that* $x = Bx + Cx$, *whenever* $\frac{\beta}{1-\alpha} < 1$.

**Remark 1.**

1.  *Note that Hypothesis* (ii) *in Theorem 5 may be replaced by "A, B, and C are weakly sequentially continuous", but the Banach algebra X must satisfy condition* $(\mathcal{P})$.
2.  *In Theorem 5, we do not require the conditions "A satisfies condition* $(\mathcal{H}1)$*" and "A*$(\Omega)$ *is relatively weakly compact", but in Theorem 3.2 in [23], these conditions are necessary.*
3.  *In Theorem 5, Condition* (i) *may be replaced by*
    $(\widetilde{ii})$ *A is regular on X and, A and C are nonlinear contractions on X with contraction functions* $\Phi_A$ *and* $\Phi_C$, *respectively, and* $L\Phi_A(r) + \Phi_C(r) < r$, *for* $r > 0$ *and* $L = \|B(\Omega)\|$.

    In the following result, we will use the notion of $\mathcal{D}$-Lipschitzian operators.

**Theorem 6.** *Let* $\Omega$ *be a nonempty, bounded, closed, and convex subset of a Banach algebra X satisfying condition* $(\mathcal{P})$ *and* $\omega$ *a subadditive MWNC on X. Let* $A, C : X \longrightarrow X$, *and* $B : \Omega \longrightarrow X$ *be three weakly sequentially continuous operators with the following conditions:*

(i)   *A is regular on X,*
(ii)  $I - \frac{I-C}{A}$ *is countably* $\alpha$-$\omega$-*contractive on* $\Omega$,
(iii) *B is countably* $\beta$-$\omega$-*contractive,*
(iv)  *A and C are* $\mathcal{D}$-*Lipschitzian with the* $\mathcal{D}$-*function* $\phi_A$ *and* $\phi_C$, *respectively, and* $L\phi_A(r) + \phi_C(r) < r$ *for* $r > 0$ *and* $L = \|B(\Omega)\|$,
(v)   $x = Ax \cdot By + Cx$, *for all* $y \in \Omega$ *implies* $x \in \Omega$.

*Then, there exists* $x \in \Omega$ *such that* $x = Ax \cdot Bx + Cx$, *whenever* $\frac{\beta}{1-\alpha} < 1$.

**Proof.** Let $y \in \Omega$ and $F_y : X \longrightarrow X$ by $F_y(x) = Ax \cdot By + Cx$.
For each $x, z \in X$, (iv) gives:

$$
\begin{aligned}
\|F_y(x) - F_y(z)\| &\leq \|Ax \cdot By - Az \cdot By\| + \|Cx - Cz\| \\
&\leq \|Ax - Az\|\|By\| + \|Cx - Cz\| \\
&\leq L\phi_A(\|x - z\|) + \phi_C(\|x - z\|).
\end{aligned}
$$

By the Boyd–Wong fixed point theorem ([24]), the mapping $F_y$ has a unique fixed point $x_y$. Hence, the operator $T = \left(\frac{I-C}{A}\right)^{-1} B : \Omega \longrightarrow X$ is well defined; and by $(v)$, we have $T(\Omega) \subset \Omega$.
Let $\{x_n\}_n$ be a sequence in $\Omega$ such that $x_n \rightharpoonup x$; as seen in the proof of Theorem 5, there exists a subsequence $\{x_{\sigma_1(n)}\}_n$ of $\{x_n\}_n$ such that $Tx_{\sigma_1(n)} \rightharpoonup y$ for some $y \in \Omega$. Since:

$$
T = AT \cdot B + CT,
$$

and $A, B$, and $C$ are weakly sequentially continuous, we obtain:

$$
Tx_{\sigma_1(n)} = A(Tx_{\sigma_1(n)}) \cdot Bx_{\sigma_1(n)} + C(Tx_{\sigma_1(n)}) \rightharpoonup y = Ay \cdot Bx + Cy
$$

Thus, $y = Tx$, and then, $T_{\sigma_1(n)} \rightharpoonup Tx$. As above, we can prove that $Tx_n \rightharpoonup Tx$; and then, $T$ is weakly sequentially continuous. By Theorems 2 and 5, $T$ is countably $\omega$-condensing.  $\square$

**Remark 2.** *Note that the hypothesis "A and C are weakly sequentially continuous" in Theorem 6 can be replaced by "$\frac{I-C}{A}$ is weakly sequentially continuous", and in this case, the condition $(\mathcal{P})$ is not required.*

**Theorem 7.** *Let $\Omega$ be a nonempty, closed, convex, and bounded subset of a Banach algebra $X$ and $\omega$ be a subadditive MWNC on $X$. Let $A, C : X \longrightarrow X$, and $B : \Omega \longrightarrow X$ be three operators satisfying the following conditions:*

*(i)    A is regular on X, and B is weakly sequentially continuous,*
*(ii)    $\frac{I-C}{A}$ is $\psi$-expansive, accretive, and continuous,*
*(iii)    $I - \frac{I-C}{A}$ is countably $\alpha$-$\omega$-contractive on $\Omega$,*
*(iv)    B is countably $\beta$-$\omega$-contractive,*
*(v)    $x = Ax \cdot By + Cx$, for all $y \in \Omega$ implies $x \in \Omega$.*

*Then, there exists $x \in \Omega$ such that $x = Ax \cdot Bx + Cx$, whenever $\frac{\beta}{1-\alpha} < 1$.*

**Proof.** For $y \in \Omega$, we define the mapping $F_y : X \longrightarrow X$ by:

$$F_y(x) = \left( I - \frac{I-C}{A} \right) x + By$$

Since $\frac{I-C}{A}$ is continuous and accretive, $I - \frac{I-C}{A}$ is continuous and pseudocontractive, and $F_y$ is continuous and pseudocontractive.

Moreover, we have:

$$\| (I - F_y)x - (I - F_y)z \| = \| \left( \frac{I-C}{A} \right) x - \left( \frac{I-C}{A} \right) z \|,$$

for all $x, z \in \Omega$, and $\frac{I-C}{A}$ is $\psi$-expansive. Then, $I - F_y$ is $\psi$-expansive, continuous, and accretive. It follows that $I - F_y$ is $m$-accretive (see [25], Corollary 3.2). By [26], Theorem 8, we deduce that $I - F_y$ is surjective. Then, there exists an $x \in X$ such that $\theta = (I - F_y)x$. It follows that:

$$x = F_y(x) = \left( I - \frac{I-C}{A} \right) x + By$$

which implies $By = \left( \frac{I-C}{A} \right) x \in \left( \frac{I-C}{A} \right)(X)$. We conclude by Theorem 5.  $\square$

In the following result, we present a nonlinear alternative of the Leary–Schauder type in Banach algebra.

**Theorem 8.** *Let $\Omega$ be a nonempty, bounded, closed, and convex subset of a Banach algebra $X$, $U$ be a weakly open subset of $\Omega$ with $\theta \in U$, and $\omega$ be a subadditive MWNC on $X$. Let $A, C : X \longrightarrow X$, and $B : \overline{U}^w \longrightarrow X$ be three operators satisfying the following conditions:*

*(i)    A is regular on X, and $\left( \frac{I-C}{A} \right)^{-1}$ exists on $B(\Omega)$,*
*(ii)    B and $\frac{I-C}{A}$ are weakly sequentially continuous,*
*(iii)    $I - \frac{I-C}{A}$ is countably $\alpha$-$\omega$-contractive on $\Omega$,*
*(iv)    B is countably $\beta$-$\omega$-contractive,*
*(v)    $x = Ax \cdot By + Cx$, for all $y \in \overline{U}^w$ implies $x \in \Omega$.*

*Then, either:*

*(i)    there exists $x \in U$ such that $x = Ax \cdot Bx + Cx$, or*
*(ii)    there exists $u \in \partial_\Omega U$ and $\lambda \in ]0, 1[$ such that $u = \lambda A \left( \frac{u}{\lambda} \right) \cdot Bu + \lambda C \left( \frac{u}{\lambda} \right)$,*

*where $\partial_\Omega U$ denotes the weak boundary of $U$ in $\Omega$ and $\frac{\alpha}{1-\beta} < 1$.*

**Proof.** Let $T := \left(\frac{I-C}{A}\right)^{-1} B$; Condition $(vi)$ implies $T(\overline{U}^w) \subset \Omega$, and $T$ is weakly sequentially continuous and countably $\omega$-condensing. Theorem 4 implies that $T$ has a fixed point in $U$, or there exists $u \in \partial_\Omega U$ and $\lambda \in ]0,1[$ such that $u = \lambda T(u)$, then either there exists $x \in U$ such that $x = Ax \cdot Bx + Cx$, or there exists $u \in \partial_\Omega U$ and $\lambda \in ]0,1[$ such that $u = \lambda A\left(\frac{u}{\lambda}\right) \cdot Bu + \lambda C\left(\frac{u}{\lambda}\right)$.　$\square$

**Corollary 2.** *Let $\Omega$ be a nonempty, bounded, closed, and convex subset of a Banach algebra $X$, $U$ be a weakly open subset of $\Omega$ with $\theta \in U$, and $\omega$ be a subadditive MWNC on $X$. Let $C : X \longrightarrow X$ and $B : \overline{U}^w \longrightarrow X$ be two operators that satisfy the following conditions:*

*(i)*　　$(I - C)^{-1}$ *exists on $B(\Omega)$,*
*(ii)*　　$B$ *and* $I - C$ *are weakly sequentially continuous,*
*(iii)*　$C$ *is countably $\alpha$-$\omega$-contractive on $\Omega$,*
*(iv)*　$B$ *is countably $\beta$-$\omega$-contractive,*
*(v)*　　$x = By + Cx$, *for all* $y \in \overline{U}^w$ *implies* $x \in \Omega$.

*Then,*

*(i)*　　*either there exists $x \in U$ such that $x = Bx + Cx$, or*
*(ii)*　*there exists $u \in \partial_\Omega U$ and $\lambda \in ]0,1[$ such that $u = \lambda Bu + \lambda C\left(\frac{u}{\lambda}\right)$,*

*where $\partial_\Omega U$ denotes the weak boundary of $U$ in $\Omega$, and $\frac{\alpha}{1-\beta} < 1$.*

**Remark 3.** *In Theorem 8, Condition $(i)$ may be replaced by*
*$(\widetilde{i})$ $\frac{I-C}{A}$ is $\psi$-expansive and $Bx \in \left(\frac{I-C}{A}\right)(X)$, for all $x \in \Omega$.*

**Theorem 9.** *Let $\Omega$ be a nonempty, closed, convex, and bounded subset of a Banach algebra $X$, $U$ be a weakly open subset of $\Omega$ with $\theta \in U$ and $\omega$ be a subadditive MWNC on $X$. Let $A, C : X \longrightarrow X$, and $B : \overline{U}^w \longrightarrow X$ be three operators satisfying the following conditions:*

*(i)*　　$A$ *is regular on $X$,*
*(ii)*　$B$ *and* $\frac{I-C}{A}$ *are weakly sequentially continuous,*
*(iii)*　$\frac{I-C}{A}$ *is $\psi$-expansive, accretive, and continuous,*
*(iv)*　$I - \frac{I-C}{A}$ *is countably $\alpha$-$\omega$-contractive on $\Omega$,*
*(v)*　　$B$ *is countably $\beta$-$\omega$-contractive,*
*(vi)*　$x = Ax \cdot By + Cx$, *for all* $y \in \overline{U}^w$ *implies* $x \in \Omega$.

*Then, either:*

*(i)*　　*there exists $x \in \Omega$ such that $x = Ax \cdot Bx + Cx$, or*
*(ii)*　*there exists $u \in \partial_\Omega U$ and $\lambda \in ]0,1[$ such that $u = \lambda A\left(\frac{1}{\lambda}u\right) \cdot Bu + C\left(\frac{1}{\lambda}u\right)$.*

*where $\partial_\Omega U$ denotes the weak boundary of $U$ in $\Omega$, and $\frac{\alpha}{1-\beta} < 1$.*

**Proof.** Define $T : \Omega \longrightarrow \Omega$ by $Tx = \left(\frac{I-C}{A}\right)^{-1} Bx$. As seen in the proof of Theorem 7, the operator $T$ is well defined; moreover, $T$ is weakly sequentially continuous and countably $\omega$-condensing, and by $(vi)$, we have $T(\overline{U}^w) \subset \Omega$; we conclude by Theorem 4.　$\square$

**Remark 4.** *If we take $A$ is the unit element in the Banach algebra $X$, we obtain Theorem 3.9 in [16].*

In the following result, the operator $\frac{I-C}{A}$ is not invertible.

**Theorem 10.** *Let $\Omega$ be a nonempty, bounded, closed, and convex subset of a Banach algebra $X$ and $\omega$ be a subadditive MWNC on $X$. Let $A, C : X \longrightarrow X$, and $B : \Omega \longrightarrow X$ be three operators that satisfy the following conditions:*

(i)    *A is regular,*

(ii)   *$I - \frac{I-C}{A}$ is countably $\alpha$-$\omega$-contractive on $\Omega$,*

(iii)  *B is countably $\beta$-$\omega$-contractive,*

(iv)   *for every net $\{x_\delta\}_\delta$, $x_\delta \in \Omega$, if $x_\delta \rightharpoonup x$, $x \in \Omega$, then, $Bx_\delta \rightharpoonup Bx$ and $\left(\frac{I-C}{A}\right) x_\delta \rightharpoonup \left(\frac{I-C}{A}\right) x$,*

(v)    *for every net $\{x_\delta\}_\delta$, $x_\delta \in \Omega$, if $\left(\frac{I-C}{A}\right) x_\delta \rightharpoonup y$, $y \in \Omega$, then there exists a weakly convergent subset of $\{x_\delta\}_\delta$,*

(vi)   *$\left(\frac{I-C}{A}\right)^{-1} Bx$ is convex, for all $x \in \Omega$;*

(vii)  *$Bx \in \left(\frac{I-C}{A}\right)(X)$ for all $x \in \Omega$ and $x = Ax \cdot By + Cx$, for all $y \in \Omega$ implies $x \in \Omega$.*

*Then, there exists $x \in \Omega$ such that $x = Ax \cdot Bx + Cx$, whenever $\frac{\beta}{1-\alpha} < 1$.*

**Proof.** By $(vii)$, the multivalued mapping:

$$
\begin{aligned}
H : \Omega &\longrightarrow P(\Omega) \\
x &\longmapsto Hx = \left(\frac{I-C}{A}\right)^{-1} Bx,
\end{aligned}
$$

is well defined.

Step 1. $H$ has a $\omega$-weakly closed graph in $\Omega \times \Omega$.

Let $\{x_\delta\}_\delta$ and $\{y_\delta\}_\delta$ be nets in $\Omega$ such that $x_\delta \rightharpoonup x \in \Omega$, $y_\delta \rightharpoonup y \in \Omega$ and $y_\delta \in Hx_\delta$.

Since $\left(\frac{I-C}{A}\right) y_\delta = Bx_\delta$, we obtain $\left(\frac{I-C}{A}\right) y_\delta \rightharpoonup \left(\frac{I-C}{A}\right) y$ and $Bx_\delta \rightharpoonup Bx$; it follows that $\left(\frac{I-C}{A}\right) y = Bx$ and then $y \in Hx$; which gives:

$$
y \in S(x,y) = \{\lambda y + (1 - \lambda)x \; : \; \lambda \in [0,1]\}
$$

then, $Hx \cap S(x,y) \neq \varnothing$, and $H$ has a $\omega$-weakly closed graph.

Step 2. By Step 1, $Hx$ is closed, for all $x \in \Omega$, and by $(vi)$, $H(\Omega) \subset P_{cl,cv}(\Omega)$.

Step 3. $H$ maps weakly compact sets into relatively weakly compact sets.

Let $K$ be a weakly compact set in $\Omega$, and let $\{y_n\}_n$ be a sequence in $H(K)$; choose $\{x_n\}_n$ in $K$ such that $y_n \in Hx_n$ for all $n \in \mathbb{N}$ and $\{x_{\sigma_1(n)}\}_n$ a subsequence of $\{x_n\}_n$ such that $x_{\sigma_1(n)} \rightharpoonup x$. By $(iv)$, $\left(\frac{I-C}{A}\right) y_{\sigma_1(n)} = Bx_{\sigma_1(n)} \rightharpoonup Bx$, and $(v)$ implies that $\{y_n\}_n$ has a weakly convergent subsequence. Then, by the Eberlein–Šmulian theorem [27], $H(K)$ is relatively weakly compact.

Step 4. $H$ is countably $\omega$-condensing.

Let $M$ be a countable subset of $\Omega$ with $\omega(M) > 0$; we have:

$$
\left(\frac{I-C}{A}\right)(Hx) = \{Bx\}, \quad \text{for all } x \in M,
$$

then, for all $y \in Hx$ we have:

$$
\left(\frac{I-C}{A}\right) y = Bx;
$$

hence:

$$
y = Bx + \left(I - \frac{I-C}{A}\right) y;
$$

consequently:

$$
Hx \subset Bx + \left(I - \frac{I-C}{A}\right)(Hx), \quad \text{for all } x \in M,
$$

then:

$$
H(M) \subset B(M) + \left(I - \frac{I-C}{A}\right)(H(M)),
$$

and:

$$\begin{aligned}
\omega(H(M)) &\leq \omega\left(B(M)\right) + \omega\left(\left(I - \frac{I-C}{A}\right)(H(M))\right) \\
&\leq \beta\omega(M) + \alpha\omega\left(H(M)\right),
\end{aligned}$$

It follows that $\omega((H(M)) \leq \frac{\beta}{1-\alpha}\omega(M) < \omega(M)$; and then, $H$ is countably $\omega$-condensing. By Theorem 3.18 in [16], we conclude that $H$ has a fixed point in $\Omega$. $\square$

The following result requires the condition "relatively weakly compact" and where $\frac{\beta}{1-\alpha} \leq 1$.

**Theorem 11.** *Let $\Omega$ be a nonempty, bounded, closed, and convex subset of a Banach algebra $X$ and $\omega$ be a positive homogenous MWNC on $X$. Let $A, C : X \longrightarrow X$, and $B : \Omega \longrightarrow X$ be three operators that satisfy the following conditions:*

(i)   *$A$ is regular on $X$, and $\left(\frac{I-C}{A}\right)^{-1}$ exists on $B(\Omega)$,*
(ii)  *$B$ and $\frac{I-C}{A}$ are weakly sequentially continuous,*
(iii) *$\left(I - \frac{I-C}{A}\right)(\Omega)$ is relatively weakly compact,*
(iv)  *$B$ is countably $\beta$-$\omega$-contractive,*
(v)   *If $\{x_n\}_n$ is a sequence in $\Omega$ such that $(I - B)x_n \rightharpoonup x$, then $\{x_n\}_n$ has a weakly convergent subsequence,*
(vi)  *$I - \frac{I-C}{A}$ is countably $\alpha$-$\omega$-contractive on $\Omega$,*
(vii) *$x = Ax \cdot By + Cx$, for all $y \in \Omega$ implies $x \in \Omega$.*

*Then, there exists $x \in \Omega$ such that $x = Ax \cdot Bx + Cx$, whenever $\frac{\beta}{1-\alpha} \leq 1$.*

**Proof.** Let $x \in \Omega$, and consider:

$$\begin{aligned}
T : \Omega &\longrightarrow \Omega \\
x &\longmapsto Tx = \left(\frac{I-C}{A}\right)^{-1} Bx;
\end{aligned}$$

by $(i)$ and $(vii)$, it is clear that $T$ is well defined.

We will show that $T$ satisfies the conditions of Theorem 3. From the proof of Theorem 5, we can see that $T$ is weakly sequentially continuous, and then, it suffices to prove that $T$ is weakly countably one-set-contractive and semi-closed at $\theta$.

Let $M$ be a countably subset of $\Omega$; we have:

$$T = B + \left(I - \frac{I-C}{A}\right)T,$$

then:

$$\begin{aligned}
\omega(T(M)) &\leq \omega(B(M)) + \omega\left(\left(I - \frac{I-C}{A}\right)(T(M))\right) \\
&\leq \beta\omega(M) + \alpha\omega(T(M)),
\end{aligned}$$

and so:

$$\omega(T(M)) \leq \frac{\beta}{1-\alpha} \leq \omega(M);$$

therefore, $T$ is weakly countably one-set-contractive.

Now, let $\{x_n\}_n$ be a sequence in $\Omega$ such that $(I - T)x_n \to \theta$.

$$y_n = (I - T)x_n = x_n - Bx_n - \left(I - \frac{I-C}{A}\right)Tx_n$$

By (*iii*), there exists a subsequence $\{x_{\sigma_1(n)}\}_n$ of $\{x_n\}_n$ such that $\left(I - \frac{I-C}{A}\right) Tx_{\sigma_1(n)} \rightharpoonup y$; and then, $(I - B)x_{\sigma_1(n)} = y_{\sigma_1(n)} + \left(I - \frac{I-C}{A}\right) Tx_{\sigma_1(n)} \rightharpoonup y$. By (*v*), we conclude that there exists a subsequence $\{x_{\sigma_1\sigma_2(n)}\}_n$ of $\{x_{\sigma_1(n)}\}_n$, which converges to some element $x$. Since $(I - T)x_{\sigma_1\sigma_2(n)} \to \theta$ and $T$ is weakly sequentially continuous, we obtain $Tx = x$, and then, $T$ is weakly semi-closed at $\theta$. □

Let $\Omega$ be a nonempty closed and convex subset of a Banach algebra $X$, and let $A, C : X \longrightarrow X$, and $B : \Omega \longrightarrow X$ be three operators. For any $D \subseteq \Omega$, we set (see [28]):

$$\mathcal{F}(A, C, B, D) = \{x \in X : x = Ax \cdot By + Cx, \ y \in D\}.$$

If $A = 1_X$ and $C = 0$, we obtain $\mathcal{F}(1_X, 0, B, D) = B(D)$.

**Theorem 12.** *Let $X$ be a Banach algebra satisfying condition $(\mathcal{P})$ and $\Omega$ be a nonempty, closed, convex, and bounded subset of $X$; $\omega$ is an MWNC on $X$. Let $A, C : X \longrightarrow X$, and $B : \Omega \longrightarrow X$ be three operators satisfying the following conditions:*

(*i*)     *$A$ is regular on $X$, and $B$ is weakly sequentially continuous,*
(*ii*)    *$I - \frac{I-C}{A}$ is a contraction on $\Omega$,*
(*iii*)   *$\omega(\mathcal{F}(A, C, B, D)) < \omega(D)$, for any countably subset $D$ of $\Omega$ with $\omega(D) > 0$,*
(*iv*)   *$\mathcal{F}(A, C, B, \Omega) \subset \Omega$,*
(*v*)    *If $\{x_n\} \subset \mathcal{F}(A, C, B, \Omega)$, then $\{Ax_n\}_n$ and $\{Cx_n\}_n$ have weakly convergent subsequences (converging respectively to $y$ and $z$), and if $x_n \rightharpoonup x$, we have $y = Ax$ and $z = Cx$.*

*Then, there exists $x \in \Omega$ such that $x = Ax \cdot Bx + Cx$.*

**Proof.** For $y \in \Omega$, we define the mapping:

$$
\begin{aligned}
F_y : X &\longrightarrow X \\
x &\longmapsto F_y(x) = \left(I - \frac{I-C}{A}\right)x + By,
\end{aligned}
$$

(*ii*) implies that $F_y$ is a contraction; then, $F_y$ has a unique fixed point $\tau(y) \in X$; we have $\tau(y) = \left(I - \frac{I-C}{A}\right)\tau(y) + By$ or equivalently $\tau(y) = A\tau(y) \cdot By + C\tau(y)$; which shows that $\tau(y) \in \mathcal{F}(A, C, B, \Omega)$. It follows that $\tau(\Omega) \subset \Omega$.

Let $M$ be a countable subset of $\Omega$ such that $\omega(M) > 0$; we have:

$$
\begin{aligned}
\tau(M) &= \{\tau(x) : x \in M\} \\
&= \{\tau(x) = A(\tau(x)) \cdot Bx + C(\tau(x)) : x \in M\} \\
&= \{\tau(x) : \tau(x) \in \mathcal{F}(A, C, B, M)\} \\
&\subseteq \mathcal{F}(A, C, B, M),
\end{aligned}
$$

Hence, $\omega(\tau(M)) \leq \omega(\mathcal{F}(A, C, B, M)) < \omega(M)$; then, $\tau$ is countably $\omega$-condensing.

Moreover, $\tau$ is weakly sequentially continuous. Indeed, let $\{x_n\}_n$ be a sequence in $\Omega$ such that $x_n \rightharpoonup x$; since $B$ is weakly sequentially continuous, we have $Bx_n \rightharpoonup Bx$, and since $\{\tau(x_n)\}_n \subset \mathcal{F}(A, C, B, \Omega)$, there exists a subsequence $\{\tau(x_{\sigma_1(n)})\}_n$ and $\{\tau(x_{\sigma_2(n)})\}_n$ of $\{\tau(x_n)\}_n$ such that $A\tau(x_{\sigma_1(n)}) \rightharpoonup y$ and $C\tau(x_{\sigma_2(n)}) \rightharpoonup z$. It follows that:

$$\tau(x_{\sigma_1\sigma_2(n)}) = A\tau(x_{\sigma_1\sigma_2(n)}) \cdot Bx_{\sigma_1\sigma_2(n)} + C\tau(x_{\sigma_1\sigma_2(n)}) \rightharpoonup y \cdot Bx + z.$$

With (*v*), we obtain $A(y \cdot Bx + z) = y$ and $C(y \cdot Bx + z) = z$; and then, $y \cdot Bx + z = A(y \cdot Bx + z) \cdot Bx + C(y \cdot Bx + z)$.

The uniqueness of the fixed point implies that $\tau(x) = y \cdot Bx + z$; and therefore, $\tau(x_{\sigma_1 \sigma_2(n)}) \rightharpoonup \tau(x)$. We claim that $\tau(x_n) \rightharpoonup \tau(x)$. For this, assume that there exists a weak neighborhood $V$ of $\tau(x)$ and a subsequence $\{x_{\varphi_1(n)}\}_n$ of $\{x_n\}_n$ such that $x_{\varphi_1(n)} \notin V$ for all $n \in \mathbb{N}$. Since $x_{\varphi_1(n)} \rightharpoonup x$, we can extract a subsequence $\{x_{\varphi_1 \varphi_2(n)}\}_n$ of $\{x_{\varphi_1(n)}\}_n$ such that $\tau(x_{\varphi_1 \varphi_2(n)}) \rightharpoonup \tau(x)$. This is not possible, since $x_{\varphi_1 \varphi_2(n)} \notin V$ for all $n \in \mathbb{N}$. We conclude that $\tau$ is weakly sequentially continuous. By Theorem 2, there exists $x \in \Omega$ such that $x = \tau(x) = Ax \cdot Bx + Cx$. $\quad\square$

If $A = 1_X$ in Theorem 12, we obtain Theorem 3.11 in [16].

**Theorem 13.** *Let $\Omega$ be a nonempty, closed, convex, and bounded subset of a Banach algebra $X$; $\omega$ is an MWNC on $X$. Let $A, C : X \longrightarrow X$, and $B : \Omega \longrightarrow X$ be three operators that satisfy the following conditions:*

*(i)      $A$ is regular on $X$, and $\frac{I-C}{A}$ is one-to-one,*
*(ii)     $I - \frac{I-C}{A}$ is nonexpansive,*
*(iii)    $B$ and $\frac{I-C}{A}$ are weakly sequentially continuous,*
*(iv)    $\omega\left(\mathcal{F}\left(A, C, B, D\right)\right) < \omega(D)$, for any countably subset $D$ of $\Omega$ with $\omega(D) > 0$,*
*(v)     $\left(I - \frac{I-C}{A}\right) x + By \in \Omega$ for all $x, y \in \Omega$,*
*(vi)    If $\{x_n\}_n \subset \Omega$ such that $\left\{\left(\frac{I-C}{A}\right) x_n\right\}_n$ is weakly convergent, then the sequence $\{x_n\}_n$ has a weakly convergent subsequence.*

*Then, there exists $x \in \Omega$ such that $x = Ax \cdot Bx + Cx$.*

**Proof.** Let $y \in \Omega$, and define $F_y : \Omega \longrightarrow X$ by:

$$F_y(x) = \left(I - \frac{I - C}{A}\right) x + By$$

By $(ii)$, $F_y$ is nonexpansive, and by $(v)$, we have $F(\Omega) \subset \Omega$. Then, by ([29], Theorem 2.15), there exists a sequence $\{x_n\}_n$ in $\Omega$ such that $\|x_n - F_y(x_n)\| \longrightarrow 0$, and then, $\left(\frac{I-C}{A}\right) x_n \longrightarrow By$. Using $(vi)$, we can extract a subsequence $\{x_{\sigma_1(n)}\}_n$ of $\{x_n\}_n$ such that $x_{\sigma_1(n)} \rightharpoonup x \in \Omega$, and then, $\left(\frac{I-C}{A}\right) x_{\sigma_1(n)} \rightharpoonup \left(\frac{I-C}{A}\right) x$; then:

$$By = \left(\frac{I - C}{A}\right) x \in \left(\frac{I - C}{A}\right) (\Omega)$$

which implies $B(\Omega) \subset \left(\frac{I-C}{A}\right)(\Omega)$.

Define $T : \Omega \longrightarrow \Omega$ by $Tx = \left(\frac{I-C}{A}\right)^{-1} Bx$. Let $D \subseteq \Omega$ and $x \in D$; the equality $Tx = A(Tx) \cdot Bx + C(Tx)$ implies that $Tx \in \mathcal{F}\left(A, C, B, D\right)$; then:

$$T(D) \subset \mathcal{F}\left(A, C, B, D\right)$$

for any subset $D$ of $\Omega$.

The assumption $(iv)$ implies that $T$ is countably $\omega$-condensing. Moreover, $T$ is weakly sequentially continuous. Indeed, let $\{x_n\}_n$ be a sequence such that $x_n \rightharpoonup x$; we have $Bx_n \rightharpoonup Bx$; then, $\left(\frac{I-C}{A}\right) Tx_n \rightharpoonup Bx$. By $(vi)$, there exists a subsequence $\{x_{\sigma_2(n)}\}_n$ such that $Tx_{\sigma_2(n)} \rightharpoonup y \in \Omega$; thus, $\left(\frac{I-C}{A}\right) Tx_{\sigma_2(n)} \rightharpoonup \left(\frac{I-C}{A}\right) y$, which leads to $\left(\frac{I-C}{A}\right) y = Bx$, and so, $Tx_{\sigma_2(n)} \rightharpoonup Tx$. As in the proof of Theorem 5, we can prove that $Tx_n \rightharpoonup Tx$, and we apply Theorem 2 to end the proof. $\quad\square$

**Remark 5.** *If we take $A = 1_X$ in Theorem 13, we obtain Theorem 3.13 in [16].*

## 4. Application

Let $X$ be a real Banach algebra satisfying condition $(\mathcal{P})$; we denote $E = C([0,1], X)$ the Banach space of all $X$-valued continuous functions defined on $[0,1]$, endowed with the norm $\|x\|_\infty = \sup_{t \in [0,1]} \|x(t)\|$. In this section, we discuss the following abstract nonlinear quadratic integral equation $((FIE))$ (see [30]):

$$x(t) = K(t, x(\xi(t))) + Tx(t) \left( q(t) + \int_0^{\sigma(t)} g(s, x(\eta(s))) ds \right), \ t \in J = [0,1],$$

where $q : J \longrightarrow X$, $g, K : J \times X \longrightarrow X$, $\xi, \sigma, \eta : J \longrightarrow J$, and $T : E \longrightarrow E$. Note that $E$ is a Banach algebra satisfying condition $(\mathcal{P})$ and the integral in $(FIE)$ is the Pettis integral, while the solutions of $(FIE)$ are in $E$. Make the following assumptions for $(FIE)$:

**Hypothesis 1** (H1).

(i)　The functions $\xi, \sigma, \eta : J \longrightarrow J$ are continuous, and $\sigma$ is nondecreasing,

(ii)　the function $q : J \longrightarrow X$ is continuous,

**Hypothesis 2** (H2).

(i)　for all $t \in [0,1]$, $K(t,.) : X \longrightarrow X$ is weakly sequentially continuous,

(ii)　for each $u \in X$, $K(., u) : J \longrightarrow X$ is continuous,

(iii)　there is a continuous function $\delta : J \longrightarrow [0, +\infty)$ with bound $\Delta = \sup_{t \in J} |\delta(t)|$ such that $\|K(t, x(t)) - K(t, y(t))\| \leq \delta(t) \|x(t) - y(t)\|$ for all $x, y \in E$ and $t \in [0,1]$,

**Hypothesis 3** (H3). *The operator* $T : E \longrightarrow E$ *satisfies:*

(i)　there is a continuous function $\gamma : J \longrightarrow [0, +\infty)$ with bound $\Gamma = \sup_{t \in J} |\gamma(t)|$ such that $\|Tx(t) - Ty(t)\| \leq \gamma(t) \|x(t) - y(t)\|$, for all $x, y \in E$ and $t \in [0,1]$,

(ii)　$T$ is weakly sequentially continuous on $E$,

(iii)　$T$ is regular on $E$; $\frac{1_E}{T}$ is well defined on $E$; $\frac{1_E}{T}$ is weakly compact; and there exists $m_0 \in [0,1)$ such that $\sup_{x \in E} \left\| 1_E - \frac{1_E}{Tx} \right\|_\infty \leq m_0$, where $1_E$ represents the unit element in the Banach algebra $E$,

**Hypothesis 4** (H4).

(i)　for each continuous $x : [0,1] \longrightarrow X$, the function $s \longmapsto g(s, x(s))$ is weakly measurable on $[0,1]$, and for almost every $t \in [0,1]$, the map $u \longmapsto g(t, u)$ is weakly sequentially continuous on $X$,

(ii)　there are a function $\phi \in L^1([0,1], \mathbb{R}^+)$ and a continuous nondecreasing function $\vartheta : [0, +\infty) \longrightarrow [0, +\infty)$ such that:

$$\|g(s, u)\| \leq \phi(s) \vartheta(\|u\|) \ a.e \text{ for all } s \in [0,1], \text{ and all } u \in X,$$

(iii)　there is a constant $0 \leq \beta < 1$ such that:

$$\mu\left(g\left([0,1] \times W\right)\right) \leq \beta \mu(W),$$

for any countably bounded subset $W$ of $X$,

**Hypothesis 5** (H5). *There is a constant* $r > 0$ *such that* $Q\Gamma + \Delta < 1$, *where:*

$$Q = Q_1 + \vartheta(r) \int_0^1 \phi(s) ds \qquad \text{with } Q_1 = \sup_{t \in J} \|q(t)\|$$

Now, we are in a position to state our main result of this section:

**Theorem 14.** *Assume that Hypotheses (H1)–(H5) hold and* $r_0 = \frac{L + Q\|T0\|_\infty}{1 - \Delta - Q\Gamma} \leq r$ *with* $L = \sup_{t \in [0,1]} \|K(t, 0)\|$; *then (FIE) has a solution in* $C([0,1], X)$ *whenever* $m_0 + \Delta(1 + m_0) < 1$ *and* $\frac{\beta}{1 - (m_0 + \Delta(1 + m_0))} < 1$.

**Proof.** The integral equation $(FIE)$ may be written in the form:

$$x(t) = Ax(t) \cdot Bx(t) + Cx(t),$$

where:

$$
\begin{aligned}
Bx(t) &= q(t) + \int_0^{\sigma(t)} g(s, x(\eta(s))) ds, \\
Ax(t) &= Tx(t), \\
Cx(t) &= K(t, x(\xi(t))).
\end{aligned}
$$

Let $\Omega = \{x \in C([0,1], X) : \|x\|_\infty \leq r_0\}$; note that $\Omega$ is a closed, convex, and bounded subset of $E$. We will show that the mappings $A$, $B$, and $C$ verify all the conditions of Theorem 6.

Step 1. We show that $A$ and $C$ are Lipschitzian. First, we verify that the mapping $C$ is well defined. Let $x \in E$, and let $\{t_n\}_n$ be a sequence in $J$ such that $t_n \to t \in J$. We have:

$$
\begin{aligned}
\|Cx(t_n) - Cx(t)\| &= \|K(t_n, x(\xi(t_n))) - K(t, x(\xi(t)))\| \\
&\leq \|K(t_n, x(\xi(t_n))) - K(t_n, x(\xi(t)))\| + \|K(t_n, x(\xi(t))) - K(t, x(\xi(t)))\| \\
&\leq \delta(t)\|x(\xi(t_n)) - x(\xi(t))\| + \|K(t_n, x(\xi(t))) - K(t, x(\xi(t)))\| \\
&\leq \Delta\|x(\xi(t_n)) - x(\xi(t))\| + \|K(t_n, x(\xi(t))) - K(t, x(\xi(t)))\|.
\end{aligned}
$$

Since $K(., x)$ is continuous and $\xi$ is continuous, then $\|Cx(t_n) - Cx(t)\| \to 0$; we conclude that $Cx \in E$. Now, let $x, y \in E$ and $t \in J$; we have:

$$\|Cx(t) - Cy(t)\| \leq \delta(t)\|x(\xi(t)) - y(\xi(t))\|,$$

then:

$$\|Cx - Cy\|_\infty \leq \Delta\|x - y\|_\infty,$$

and we have:

$$
\begin{aligned}
\|Ax(t) - Ay(t)\| &= \|Tx(t) - Ty(t)\| \\
&\leq \gamma(t)\|x(t) - y(t)\|,
\end{aligned}
$$

then:

$$\|Ax - Ay\|_\infty \leq \Gamma\|x - y\|_\infty.$$

Thus, $A$ and $C$ are Lipschitzians with the Lipschitz constants $\Delta$ and $\Gamma$, respectively.

Step 2. From the assumption (H3)$(ii)$, the mapping $A$ is weakly sequentially continuous on $E$. Now, we show that $C$ is weakly sequentially continuous on $E$; for this, let $\{x_n\}_n$ in $E$ such that $x_n \rightharpoonup x \in E$, then $\{x_n\}_n$ is bounded on $E$; from Dobrokov's theorem ([31], p. 36), we get for all $t \in [0,1]$, $x_n(t) \rightharpoonup x(t)$. Since $K(t, .)$ is weakly sequentially continuous for all $t \in [0,1]$, we get $Cx_n(t) \rightharpoonup Cx(t)$. Again, from Dobrokov's theorem, we deduce that $Cx_n \rightharpoonup Cx$, then $C$ is weakly sequentially continuous on $E$. Now, we prove that $B$ is weakly sequentially continuous. Firstly, we verify that if $x \in \Omega$, then $Bx \in E$. Let $x \in \Omega$ and $t, t' \in [0,1]$, such that $t \leq t'$; without loss of generality, we may assume that $Bx(t) - Bx(t') \neq 0$. Using the Hahn–Banach

theorem, we get that there exists $x^* \in X^*$ such that $x^*(Bx(t) - Bx(t')) = \|Bx(t) - Bx(t')\|$ and $\|x^*\|_* = 1$; hence:

$$
\begin{aligned}
\|Bx(t) - Bx(t')\| &= x^*(Bx(t) - Bx(t')) \\
&= x^*(q(t) - q(t')) + \int_{\sigma(t)}^{\sigma(t')} x^*(g(s, x(\eta(s)))) ds \\
&\leq \sup_{t \in J} \|q(t) - q(t')\| + \vartheta(\|x\|_\infty) \int_{\sigma(t)}^{\sigma(t')} \phi(s) ds \\
&\leq \sup_{t \in J} \|q(t) - q(t')\| + \vartheta(r_0) \int_{\sigma(t)}^{\sigma(t')} \phi(s) ds;
\end{aligned}
$$

consequently, $Bx \in E$. As $q$ and $\sigma$ are uniformly continuous on the compact $[0,1]$, we get that $B(\Omega)$ is an equicontinuous family of functions. Now, we show that $B$ is weakly sequentially continuous on $\Omega$. Let $\{x_n\}_n$ be a sequence in $\Omega$ such that $x_n \rightharpoonup x \in \Omega$, then we get for all $t \in [0,1]$, $x_n(t) \rightharpoonup x(t)$. Furthermore, for $n \in \mathbb{N}$ and $x^* \in X^*$:

$$
x^*(Bx_n(t)) = x^*(q(t)) + \int_0^{\sigma(t)} x^*(g(s, x_n(\eta(s)))) ds, \quad \text{for all } t \in J,
$$

From (H1)$(i)$ and (H4)$(i)$, we have $x^*(g(s, x_n(\eta(s)))) \to x^*(g(s, x(\eta(s))))$ for all $s \in [0,1]$. The Lebesgue dominated convergence theorem yields:

$$
\int_0^{\sigma(t)} x^*(g(s, x_n(\eta(s)))) ds \longrightarrow \int_0^{\sigma(t)} x^*(g(s, x(\eta(s)))) ds,
$$

then $Bx_n(t) \rightharpoonup Bx(t)$; by Dobrokov's theorem ([31], p. 36), we get $Bx_n \rightharpoonup Bx$.

Step 3. $B$ i countably $\beta$-$\omega$-contractive. First, we show that $B(\Omega)$ is bounded. Let $x \in \Omega$ and $t \in [0,1]$. Without loss of generality, we may assume that $Bx(t) \neq 0$. Using the Hahn–Banach theorem, we deduce that there exists $x^* \in X^*$ such that $x^*(Bx(t)) = \|Bx(t)\|$ and $\|x^*\|_* = 1$. Hence,

$$
\begin{aligned}
\|Bx(t)\| &= x^*(Bx(t)) \\
&= x^*(q(t)) + \int_0^{\sigma(t)} x^*(g(s, x(\eta(s)))) ds \\
&\leq \sup_{t \in J} \|q(t)\| + \int_0^1 \|g(s, x(\eta(s)))\| ds \\
&\leq Q_1 + \vartheta(r) \int_0^1 \phi(s) ds = Q,
\end{aligned}
$$

then $B(\Omega)$ is bounded.

Now, let $V$ be a countably bounded subset of $\Omega$; for each $t \in [0,1]$, we have by ([32], Theorem 3):

$$
\begin{aligned}
\mu(B(V)(t)) &\leq \mu\left(\left\{\int_0^{\sigma(t)} g(s, x(\eta(s))) ds : x \in V\right\}\right) \\
&\leq \mu\left(\sigma(t)\overline{co}\{g(s, x(\eta(s))) : x \in V, s \in [0,1]\}\right) \\
&\leq \mu\left(g([0,1] \times V([0,1]))\right) \\
&\leq \beta\mu\left(V([0,1])\right) \\
&\leq \beta \sup_{t \in J} \mu(V(t)) \\
&\leq \beta\mu(V),
\end{aligned}
$$

because $V$ is bounded, then we can apply Theorem 1.

Since $V(B)$ is bounded and equicontinuous, again, by Theorem 1, we get:

$$\mu(B(V)) \leq \beta\mu(V).$$

Consequently, $B$ is countably $\beta$-$\mu$-contractive.

Step 4. Now, we prove that $I - \frac{I-C}{A}$ is countably $\alpha'$-$\mu$-contractive where $\alpha' = m_0 + \Delta(1 + m_0)$. Firstly, for all $x \in E$ by Step 1, we have $Cx \in E$, and by (H3)(iii), we have $\frac{1_E}{Ax} \in E$; hence, $\left(I - \frac{I-C}{A}\right)x \in E$. Now, let $x \in \Omega$; we have:

$$
\begin{aligned}
\left\|\left(I - \frac{I-C}{A}\right)x\right\|_\infty &= \left\|x - \frac{x - Cx}{Ax}\right\|_\infty \\
&\leq \left\|1_E - \frac{1_E}{Ax}\right\|_\infty \|x\|_\infty + \left\|\frac{1_E}{Ax}\right\|_\infty \|Cx\|_\infty \\
&\leq m_0 r_0 + (1 + m_0)(\Delta r_0 + L),
\end{aligned}
$$

then $\left(I - \frac{I-C}{A}\right)(\Omega)$ is bounded. Now, let $V$ be a bounded subset of $\Omega$ such that $\mu(V) > 0$; note that for all $x \in V$, we have:

$$
\begin{aligned}
\left(I - \frac{I-C}{A}\right)x &= x - \frac{x - Cx}{Ax} \\
&= \left(1_E - \frac{1_E}{Ax}\right) \cdot x + \frac{1_E}{Ax} \cdot Cx,
\end{aligned}
$$

then:

$$\left(I - \frac{I-C}{A}\right)(V) \subset \left(1_E - \frac{1_E}{A(V)}\right) \cdot V + \frac{1_E}{A(V)} \cdot C(V),$$

because $\frac{1_E}{A}$ is weakly compact; then, by the assumption (H3)(iii), we get:

$$
\begin{aligned}
\mu\left(\left(I - \frac{I-C}{A}\right)(V)\right) &\leq \mu\left(\left(1_E - \frac{1_E}{A(V)}\right) \cdot V\right) + \mu\left(\frac{1_E}{A(V)} \cdot C(V)\right) \\
&\leq \left\|1_E - \frac{1_E}{A(V)}\right\|_\infty \mu(V) + \left\|\frac{1_E}{A(V)}\right\|_\infty \mu(C(V)).
\end{aligned}
$$

because $C$ is $\Delta$-Lipschitzian and weakly sequentially continuous; by Lemma 1, we get $\mu(C(V)) \leq \Delta\mu(V)$, then

$$\mu\left(\left(I - \frac{I-C}{A}\right)(V)\right) \leq \left\|1_E - \frac{1_E}{A(V)}\right\|_\infty \mu(V) + \left\|\frac{1_E}{A(V)}\right\|_\infty \Delta\mu(V),$$

then:

$$
\begin{aligned}
\mu\left(\left(I - \frac{I-C}{A}\right)(V)\right) &\leq m_0\mu(V) + (1 + m_0)\Delta\mu(V) \\
&\leq \alpha'\mu(V),
\end{aligned}
$$

where $\alpha' = m_0 + \Delta(1 + m_0) < 1$; then, $I - \frac{I-C}{A}$ is countably $\alpha'$-$\mu$-contractive.

Step 5. We show that for all $x \in E$ and $y \in \Omega$, if $x = Ax \cdot By + Cx$, then $x \in \Omega$. We have for all $t \in [0, 1]$:

$$x(t) = Ax(t) \cdot By(t) + Cx(t),$$

then,

$$
\begin{aligned}
\|x(t)\| &\leq \|Cx(t)\| + \|Ax(t) \cdot By(t)\| \\
&\leq \|Cx(t)\| + \|Ax(t)\|\|By(t)\| \\
&\leq \Delta\|x(t)\| + L + Q\left(\Gamma\|x(t)\| + \|A0\|\right),
\end{aligned}
$$

then,

$$
\|x(t)\| \leq \frac{L + Q\|A0\|}{1 - \Delta - Q\Gamma} = r_0,
$$

hence,

$$
\|x\|_\infty \leq r_0,
$$

consequently, $x \in \Omega$.

Applying Theorem 6, we get a fixed point for $A \cdot B + C$ and hence a solution to $(FIE)$ in $E$. □

## 5. Example

Consider the Banach algebra $E = \mathcal{C}([0,1], \mathbb{R})$ of all continuous real-valued functions on $J = [0,1]$, with norm $\|x\|_\infty = \sup_{t \in [0,1]} |x(t)|$. In this case, $X = \mathbb{R}$, and $E$ is a Banach algebra satisfying condition $(\mathcal{P})$ and reflexive. Let $b : [0,1] \longrightarrow X$ be a continuous and nonnegative function such that $\sup_{t \in J} |b(t)| = \frac{1}{4}$. We consider the following nonlinear integral equation:

$$
x(t) = \frac{1}{4}t^3 x\left(\frac{t^2}{2}\right) + \left(1 + \int_0^t \frac{b(s)}{1 + |x(s)|} ds\right) \cdot \left(\sqrt{t} + \int_0^t \frac{s^2}{20} \frac{|x(s)| \cdot x(s)}{e^{|x(s)|}} ds\right), \quad t \in J. \tag{3}
$$

To show that (3) has a solution in $E$, we will verify that all conditions of Theorem 14 are satisfied.

Define $K : [0,1] \times \mathbb{R} \longrightarrow \mathbb{R}$, by $K(t, x(t)) = \frac{1}{4}t^3 x\left(\frac{t^2}{2}\right)$ (in this case $\xi(t) = \frac{t^2}{2}$). For all $t \in [0,1]$, the function $K(t,.) : X \longrightarrow X$ is continuous (then weakly sequentially continuous, because $X = \mathbb{R}$), and for all $x \in X$, the function $K(.,x) : J \longrightarrow X$ is continuous. Now, let $x, y \in E$ and $t \in [0,1]$; we have:

$$
|K(t, x(t)) - K(t, y(t))| \leq \delta(t)|x(t) - y(t)|
$$

where the function $\delta : t \mapsto \frac{1}{4}t^3$ is continuous with bound $\Delta = \sup_{t \in J} |\delta(t)| = \frac{1}{4}$.

Next, we introduce the function $T : E \longrightarrow E$ such that $Tx(t) = 1 + \int_0^t \frac{b(s)}{1 + |x(s)|} ds$ for all $t \in J$. As seen in Step 2 in the proof of Theorem 14, the operator $T$ is weakly sequentially continuous, regular on $E$, and $\frac{1_E}{T}$ is well defined on $E$. Let $x \in E$ and $t \in [0,1]$; we have:

$$
\left|1 - \frac{1}{Tx(t)}\right| = \frac{\int_0^t \frac{b(s)}{1 + |x(s)|} ds}{1 + \int_0^t \frac{b(s)}{1 + |x(s)|} ds} \leq \int_0^1 b(s)\, ds \leq \frac{1}{4};
$$

thus, $\sup_{x \in X} \|1_E - \frac{1_E}{Tx}\|_\infty \leq m_0$, where $m_o = \frac{1}{4}$.

Moreover, $\frac{1_E}{T}$ is weakly compact on $E$; indeed, let $x \in E$, and let $t, t' \in J$ such that $t \leq t'$. Without loss of generality, we may assume that $\left(\frac{1_E}{T}\right)x(t) - \left(\frac{1_E}{T}\right)x(t') \neq 0$. Using the Hahn–Banach theorem, we deduce that there exists $x^* \in X^*$ such that $x^*\left(\left(\frac{1_E}{T}\right)x(t) - \left(\frac{1_E}{T}\right)x(t')\right) = \left|\left(\frac{1_E}{T}\right)x(t) - \left(\frac{1_E}{T}\right)x(t')\right|$ and $\|x^*\|_* = 1$, hence,

$$
\left|\left(\frac{1_E}{T}\right)x(t) - \left(\frac{1_E}{T}\right)x(t')\right| \leq \frac{1}{4}|t - t'|,
$$

then $\left(\frac{1_E}{T}\right)(E)$ is weakly equicontinuous. Now, let $\{x_n\}_n$ be a sequence in $E$, and fix $t \in J$; we have:

$$\left|\left(\frac{1_E}{T}\right)x_n(t)\right| = \left|\frac{1}{1 + \int_0^t \frac{b(s)}{1+|x_n(t)|}ds}\right| \leq 1;$$

therefore, $\{\left(\frac{1_E}{T}\right)x_n(t)\}_n$ is weakly equi-bounded. Let $t \in J$; since $X = \mathbb{R}$ is reflexive, then by [33], the set $\{\left(\frac{1_E}{T}\right)x_n(t) : n \in \mathbb{N}\}$ is weakly relatively sequentially compact. The Arzela–Ascoli theorem implies that there exists a subsequence $\{\left(\frac{1_E}{T}\right)x_{\sigma(n)}\}_n$ such that $\left(\frac{1_E}{T}\right)x_{\sigma(n)} \rightharpoonup \left(\frac{1_E}{T}\right)x \in E$; then, $\left(\frac{1_E}{T}\right)(E)$ is relatively weakly compact. Therefore, $\frac{1_E}{T}$ is weakly compact.

Let $x, y \in E$ and $t \in [0, 1]$; we have:

$$\begin{aligned}|Tx(t) - Ty(t)| &\leq \int_0^t b(s)(|x(s) - y(s)|)ds \\ &\leq \gamma(t)\|x - y\|_\infty,\end{aligned}$$

where $\gamma : t \mapsto \frac{t}{4}$ is continuous with bound $\Gamma = \sup_{t \in J}|\gamma(t)| = \frac{1}{4}$.

Finally, we define $g : [0, 1] \times X \longrightarrow X$, by $g(s, x(s)) = \frac{s^2}{20}\frac{|x(s)| \cdot x(s)}{e^{|x(s)|}}$. For each $u \in X$, the function $g(., u) : [0, 1] \longrightarrow X$ is weakly measurable on $[0, 1]$, and for almost every $t \in [0, 1]$, the function $g(t, .) : X \longrightarrow X$ is continuous (then weakly sequentially continuous). Furthermore, we have:

$$|g(s, u)| \leq \vartheta(|u|)\phi(s) \ a.e \text{ for all } s \in [0, 1], \text{ and all } u \in X,$$

where $\phi(s) = s^2$ and $\vartheta(v) = \frac{v}{20}$ for all $v \in [0, +\infty)$ since $e^{|z|} \geq |z|$ for all $z \in X$.
Moreover, if $W$ is a countably bounded subset of $X$, we have:

$$\begin{aligned}\mu\left(g([0, 1] \times W)\right) &= \mu\left(\{g(s, u) : u \in W \text{ and } s \in [0, 1]\}\right) \\ &\leq \mu\left(\left\{\left(\frac{1}{20}\frac{s^2.|u|}{e^{|u|}}\right).u : u \in W \text{ and } s \in [0, 1]\right\}\right) \\ &\leq \mu\left([0, \frac{1}{5}].W\right) \\ &\leq \frac{1}{5}\mu(W),\end{aligned}$$

Then:

$$\mu(g([0, 1] \times W)) \leq \beta\mu(W), \text{ where } \beta = \frac{1}{5}.$$

We set $q : [0, 1] \longrightarrow [0, +\infty)$, such that $q(t) = \sqrt{t}$; we have that $q$ is continuous and $Q_1 = \sup_{t \in J}|q(t)| = 1$.

If we take $r = 4$, we get $Q = Q_1 + \vartheta(4)\int_0^1 \phi(s)ds = \frac{16}{15}$ and $Q\Gamma + \Delta = \frac{31}{60} < 1$ (then, for all $s \in \mathbb{R}^+$, $Q\phi_A(s) + \phi_C(s) = Q\Gamma s + \Delta s < s$ where $\phi_A(s) = \Gamma s$ and $\phi_C(s) = \Delta s$).
Now, we have $\|T0\|_\infty = \sup_{t \in J}|1 + \int_0^t b(s) ds| \leq \frac{5}{4}$ and $r_0 = \frac{L + Q\|T0\|_\infty}{1 - \Delta - Q\Gamma} \leq \frac{80}{29}$, then $r_0 \leq r$, $m_0 + (1 + m_0) \cdot \Delta = \frac{9}{16} < 1$ and $\frac{\beta}{1 - (m_o + (1 + m_o) \cdot \Delta)} = \frac{16}{35} < 1$.

Theorem 14 proves the existence of a solution to Equation (3).

## 6. Conclusions

In this paper, we proved some fixed point theorems for the nonlinear operator $A \cdot B + C$ in a Banach algebra under a weak topology and with the help of the measure of weak noncompactness. Our results improved and generalized some interesting fixed point theorems in the literature. Our examples

showed that the results in this paper can be applied to prove the existence of the solution of a nonlinear integral equation in Banach algebra.

**Author Contributions:** All authors contributed equally and significantly to writing this article.

**Funding:** We have no funding for this article.

**Acknowledgments:** The authors are thankful to the Editors and the anonymous referees for their valuable comments, which reasonably improved the presentation of the manuscript.

**Conflicts of Interest:** The authors declare that they have no competing interests.

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
