# Peer review of "Measure of Weak Noncompactness and Fixed Point Theorems in Banach Algebras with Applications"

_axioms, doi:10.3390/axioms8040130_

Round 1

Reviewer 1 Report

    The authors prove some fixed point theorems for the nonlinear operator in Banach algebra under the conditions of weak topology and measure of weak noncompactness. These results are new and interesting. The proof seems to be correct.

  By the above comments, we will recommend the paper for publication in Axioms

Author Response

Thank for your comments

The linguistic aspect has been reviewed and improved.

The introduction was developped and reformulated

The general presentation of the manuscript has been improved

Reviewer 2 Report

Dear Authors:       
     I have read your paper carefully.   In this paper, the authors  proved some fixed point theorems for the nonlinear operator in Banach algebra and give some examples to illustrate the correctness of their results. Although the conclusions are right ,  in my opinion, the significance  and quality of this paper is not enough, especially in English writing, there are many problems. For example, the first sentence of introduction (Line 8-10) and line 28-35 and so on .  Hence, I think the authors need extensive editing of English language and style required.
      From a linguistic point of view, I don't think this paper has reached the level of publication. 
     Thank you.

Author Response

Dear Reviewer

Thank you for your comments

The linguistic aspect has been reviewed and improved. The general presentation of the manuscript has been improved

The introduction has been improved and developped to provide a historical overview of the developpment of this topic and to cite the most relevant references.

The manuscript structure has been improved and the proofs revised and improved

A conclusion was added. 

Reviewer 3 Report

In this manuscript, the four authors proved some fixed point theorems on Banach algebras. The fixed point results presented in this manuscript are obtained under weak topology and measure of weak noncompactness. An example of an application of the main results to a nonlinear integral equation in Banach algebras is also presented. However, the introduction section is too short. More motivation, background and history of this research filed should be addressed in Section 1. Some corollaries should be given as the sub-results of the main results. A concluding remark is needed to highlight this manuscript. The references are not sound. Some latest related published papers should be included into the References to support this manuscript, such as, [F.N.Sarvestani, S.M. Vaezpour, M. Asadi, A characterization of the generalized KKM mappings via the measure of noncompactness in complete geodesic spaces, J. Nonlinear Funct. Anal. 2017 (2017), Article ID 8], [X. Qin, N.T. An, Smoothing algorithms for computing the projection onto a Minkowski sum of convex sets, Comput. Optim. Appl. (2019). https://doi.org/10.1007/s10589-019-00124-7], [C.D. Alecsa, A. Petrusel, On some fixed point theorems for multi-valued operators by altering distance technique, J. Nonlinear Var. Anal. 1 (2017), 237-251] and [X. Qin, J.C. Yao, A viscosity iterative method for a split feasibility problem, J. Nonlinear Convex Anal. 20 (2019), 1497-1506].  

Author Response

Dear Reviewer

Thank you for your comments.

In response to your comments and suggestions, we inform you that:

The introduction has been improved and developped in order to provide a historical overview  of the developpment of this topic and to refer to the most relevant publications. We added  a corollary for theorem5 and a corollary for theorem 8. A conclusion is added in the manuscript Some relevant references are added in the bibliography The manuscript presentation is also improved The linguistic aspect has been reviewed and improved.